# A feedback amplification loop between stem cells and their progeny promotes tissue regeneration and tumorigenesis

Jun Chen[1,2], Na Xu[1], Huanwei Huang[1], Tao Cai[1], Rongwen Xi[1,2]*

[1]National Institute of Biological Sciences, Beijing, China; [2]Graduate School of Peking Union Medical College, Beijing, China

**Abstract** Homeostatic renewal of many adult tissues requires balanced self-renewal and differentiation of local stem cells, but the underlying mechanisms are poorly understood. Here we identified a novel feedback mechanism in controlling intestinal regeneration and tumorigenesis in *Drosophila*. Sox21a, a group B Sox protein, is preferentially expressed in the committed progenitor named enteroblast (EB) to promote enterocyte differentiation. In *Sox21a* mutants, EBs do not divide, but cannot differentiate properly and have increased expression of mitogens, which then act as paracrine signals to promote intestinal stem cell (ISC) proliferation. This leads to a feedback amplification loop for rapid production of differentiation-defective EBs and tumorigenesis. Notably, in normal intestine following damage, Sox21a is temporally downregulated in EBs to allow the activation of the ISC-EB amplification loop for epithelial repair. We propose that executing a feedback amplification loop between stem cells and their progeny could be a common mechanism underlying tissue regeneration and tumorigenesis.

*For correspondence: xirongwen@nibs.ac.cn

## Introduction

Adult stem cells have important roles in maintaining tissue and organ homeostasis by their prolonged ability to produce progenitor cells that differentiate into multiple types of mature cells. Production of the progenitor cells from stem cells must be coordinated with cell differentiation and the overall tissue demand, as disruption of this coordination could lead to tissue degeneration, if cell production is not sufficient, or hyperplasia/ tumorigenesis, if the cell production is unrestricted and exceeds the pace of cell differentiation. However, the molecular mechanism that coordinates progenitor cell proliferation with cell differentiation is largely unknown.

The adult *Drosophila* midgut has been established as a simple and useful system for the study of the stem cell behavior during homeostatic tissue renewal and in response to environmental changes (*Biteau et al., 2011*; *Jiang and Edgar, 2012*). Like mammalian intestine, the *Drosophila* midgut epithelium is constantly replenished by adult intestinal stem cells (ISCs) (*Micchelli and Perrimon, 2006*; *Ohlstein and Spradling, 2006*), although at a relatively slower pace. In addition, signaling pathways that regulate mammalian ISC activity, such as Wnt, JAK/STAT, EGFR/Ras, Hippo, BMP and Notch, also play important roles in regulating *Drosophila* ISC activity during normal homeostasis and/or stress conditions (reviewed by) (*Biteau et al., 2011*; *Pasco et al., 2015*). The *Drosophila* ISC, which generates a relatively simple stem cell lineage, can be specifically marked by Delta (Dl), the Notch ligand. After each asymmetric division, an ISC will produce a new ISC and a committed progenitor cell named enteroblast (EB), which will further differentiate into either an enterocyte or an enteroendocrine cell, depending on the levels of Notch activation it received from ISCs (*Ohlstein and Spradling, 2007*). Enterocyte differentiation from EB requires high levels of Notch activation, and JAK/STAT signaling activity is required for both enterocyte and enteroendocrine cell differentiation from

**eLife digest** Within our bodies we have stem cells that are responsible for maintaining many of our tissues in a healthy state and healing wounds after an injury. When these adult stem cells divide, they can produce daughter cells. Through a process called differentiation, these daughter cells can become mature cells that replace old or damaged cells. However, the stem cells also produce copies of themselves – in a process known as self-renewal – to ensure that an organ does not run out of stem cells. The rates of differentiation and self-renewal must be carefully balanced: too much differentiation can eventually lead to the degeneration of the tissue, whereas too much self-renewal can cause tumors to develop.

One of the main questions in the stem cell field is how tissues and organs balance these opposing processes. The fruit fly mid-gut is a model system for investigating this question, and is similar to the intestine of mammals. The mid-gut is composed of three main cell types: intestinal stem cells, enteroblasts (immediate daughter cells of the stem cells) and enterocytes (fully differentiated, mature cells).

Previously published data showed that a protein called Sox21a is present in the intestinal stem cells and enteroblasts, but not in the mature enterocytes. To investigate the role of Sox21a in more detail, Chen et al. deleted the gene that produces Sox21a in fruit flies. These mutant flies developed tumors in their guts, indicating that Sox21a is a tumor suppressor. Further experiments revealed that Sox21a normally drives enteroblasts to differentiate into enterocytes, and also prevents the enteroblasts from communicating with the stem cells to indicate that more enteroblasts are needed.

In further experiments, Chen et al. gave otherwise healthy fruit flies a drug that injured their guts. This caused Sox21a activity to decrease temporarily, allowing more enteroblasts to be produced from intestinal stem cells to repair the damage. The mid-gut therefore has an intricate "feedback amplification" system that maintains an appropriate number of each type of cell.

In future, other experiments will be needed to determine whether similar feedback amplification systems are found in other tissues, and to investigate the extent to which these systems are found in mammals. Furthermore, understanding this process in more depth could increase our knowledge about how cancerous tumors grow.

---

EB (*Beebe et al., 2010*; *Jiang et al., 2009*; *Lin et al., 2010*). Aside from the signaling pathways, many transcription factors have been identified as important regulators of cell differentiation. Enterocyte differentiation from EB requires downregulation of Escargot (Esg) and activation of Pdm1 (*Korzelius et al., 2014*; *Loza-Coll et al., 2014*), whereas enteroendocrine cell differentiation from EB requires release of the inhibition by the transcriptional repressor Tramtrack and activation of acheate-scute complex (AS-C) genes and Prospero (Pros), the enteroendocrine cell determination factor (*Bardin et al., 2010*; *Wang et al., 2015*; *Zeng and Hou, 2015*). It is largely unclear how these signaling pathways and transcription factors are coordinately regulated for balanced self-renewal of ISCs and differentiation of EBs to maintain intestinal homeostasis.

Sox family transcription factors, which share a DNA binding high-mobility-group domain, are known as important regulators of cell fate decisions during development and in adult tissue homeostasis (*Kamachi and Kondoh, 2013*; *Sarkar and Hochedlinger, 2013*). In mouse small intestine, Sox2 is expressed in ISCs and progenitor cells and is critical for ISC maintenance and differentiation of Paneth cells (*Furuyama et al., 2011*; *Sato et al., 2011*). Several Sox family proteins have been identified in *Drosophila* (*McKimmie et al., 2005*), but their potential roles in the ISC lineage are unclear. Here we characterized the function of a *Drosophila* Sox gene, *Sox21a*, in the ISC lineage. We find that Sox21a is expressed in EBs and acts as a tumor suppressor in the midgut epithelium. One important function of Sox21a is to promote enterocyte differentiation by inducing Pdm1 expression. By studying its tumor suppressing function, we identified a novel feedback amplification loop between ISC and EB, which is normally suppressed by Sox21a. Temporal activation of this loop is essential for damage-induced intestinal regeneration, whereas sustained activation of this loop leads to tumorigenesis. Therefore, we revealed a novel mechanism that coordinates stem cell activity with progenitor cell differentiation, and connects regeneration with tumorigenesis.

## Results

### Sox21a is preferentially expressed in differentiating EBs

From cell-type specific gene expression analysis of *Drosophila* midgut cells (*Dutta et al., 2015* and unpublished data), we noticed a Sox family gene, *Sox21a*, whose RNA expression was detected only in ISCs and EBs. Interestingly, *Sox21a* seems to be mainly expressed in midgut, but not other organs in larva and adult *Drosophila* (*Chintapalli et al., 2007*). To characterize its expression pattern in vivo, we first generated polyclonal antibodies against Sox21a, and demonstrated that this antisera could specifically mark Sox21a antigen in the midgut epithelium (*Figure 1E* and *Figure 1—figure supplement 1*). Immunostaining of the wild type midgut with this antisera revealed that Sox21a was largely undetectable in the midgut of newly eclosed and young flies of two to three days old (*Figure 1A*). Its expression began to appear with age and at 4–5 days old, weak Sox21a expression appeared specifically in Dl$^+$ ISCs and Notch-activated EBs that can be marked by a Notch activation reporter, Su(H)Gbe>GFP (NRE>GFP) (*Figure 1B*). At 7 days old, its expression could also be detected in early ECs, which display increased cell ploidy (*Figure 1C–D*). The Dl$^+$ ISC and its immediate daughter EB (marked by NRE>GFP) are usually adjacent to each other, forming an ISC-EB pair (*Ohlstein and Spradling, 2007*). In each ISC-EB pair, the level of Sox21a expression was usually higher in EB than in ISC (*Figure 1C,F*). In each progenitor cell nest where the early EC still retained NRE>GFP expression, the early EC usually displayed a higher Sox21a expression level than the EB or the ISC in the same ISC-EB-early-EC cell nest (*Figure 1D,F*), suggesting that Sox21a is up-regulated in differentiating EBs. Consistent with this notion, A GFP reporter driven by an enhancer Gal4 line for *Sox21a* (GMR43E09-Gal4) showed specific expression of GFP in ISCs and EBs. Again, the intensity of GFP was much higher in EBs than in ISCs (*Figure 1G*). Previous studies suggest that EB differentiation requires mesenchymal-epithelial transition through downregulation of Esg (*Antonello et al., 2015*; *Korzelius et al., 2014*). Interestingly, in fixed midgut where Sox21a was not expressed, EBs commonly displayed triangle-shaped morphology (*Figure 1A*). We propose that these EBs are probably the dormant EBs that remain in the undifferentiated state (*Figure 1H*). By contrast, in EBs where Sox21a began to be expressed, EBs instead displayed oval-shaped morphology (*Figure 1C,D*). These EBs are likely the activated EBs that are undergoing mesenchymal-epithelial transition (*Figure 1H*). Taken together, these observations suggest that Sox21a is preferentially expressed in differentiating EBs in the intestinal epithelium. This unique expression pattern indicates that Sox21a might have a role during the process of EB differentiation into enterocyte.

### *Sox21a* null mutants develop intestinal tumors composed of differentiation-defective cells

To study the function of *Sox21a*, we generated two gene deletion lines using the CRISPR/Cas9 system, and named these two alleles as *Sox21a [JC1]* and *Sox21a [JC2]*, which carries approximately 2.15 kb and 2.65 kb DNA fragment deletion, respectively, in the gene locus (*Figure 2A*). Because the coding region of *Sox21a* is almost entirely deleted in both alleles, they can be regarded as null alleles for *Sox21a*. Consistent with the midgut-restricted expression during development, *Sox21a* homozygous mutant flies are viable and fertile. Staining midgut of newly eclosed mutants showed that the general gut morphology and the cellular makeup of the monolayer midgut epithelium was largely normal, although the epithelial cells appeared to be more sparsely distributed in the epithelium compared to the wild type intestinal epithelium (*Figure 2B*). This indicates that Sox21a could have a role in regulating the proliferation of midgut progenitor cells during development. Strikingly, when the mutant flies got older, multilayered cell clusters composed of 50 or more cells (termed tumors hereafter) were frequently observed in both anterior and posterior midgut, and tumor incidence steadily increased with age (*Figure 2C,D*). At day 6 after eclosion, approximately 49.1% and 57.9% of [JC1] and [JC2] homozygous female flies had at least one tumor in the midgut, respectively, and at day 11, approximately 77.3% and 94.1% of [JC1] and [JC2] flies had gut tumors, respectively (*Figure 2D*). Tumor size also grew with time, and in aged mutants, the tumor masses often filled up the entire lumen space, suggesting that tumors grow rapidly once it is initiated.

To study the cellular outcome of the mutant cells, we examined the mutant midgut with several cell fate markers at day 2 and day 7 after eclosion. Like all other cells, the Su(H)Gbe-lacZ (NRE-lacZ)$^+$ EBs are more sparsely distributed in the epithelium of the mutant intestine than of the wild type

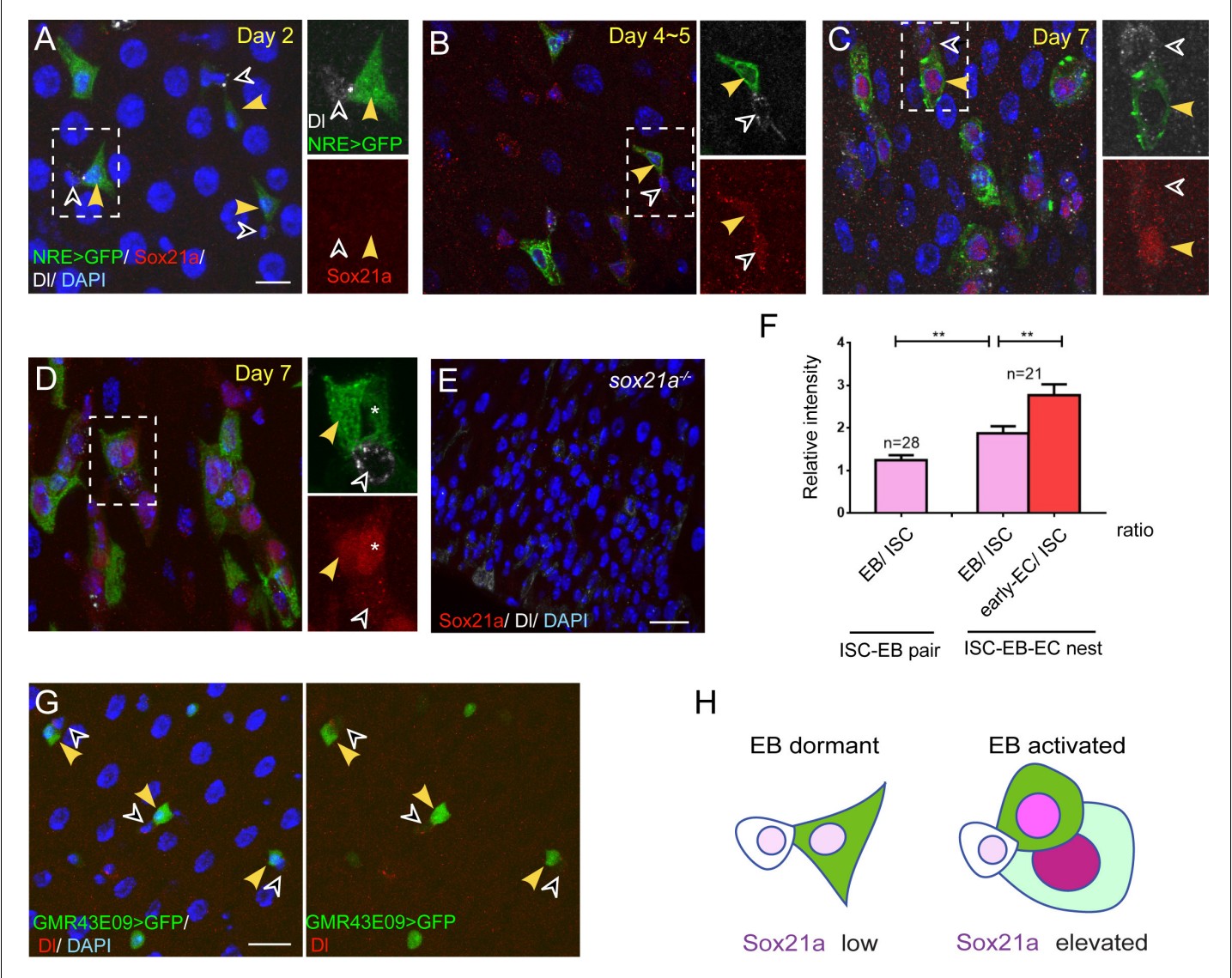

**Figure 1.** Sox21a is preferentially expressed in differentiating EBs. (A–D) Sox21a (red) expression in midguts of 2 to 7 days old females co-stained with Dl (white) and NRE>GFP (green). Separate color channels for the insets were enlarged and displayed on the right side of each corresponding images. ISCs and EBs were indicated by white or yellow arrowheads, respectively. Sox21a expression was largely undetectable in the midgut of newly eclosed females (A) Weak Sox21a expression was detectable in ISCs and EBs in midgut of 5 days old females (B). Sox21a expression increased with age, but EB showed a relatively higher expression level than ISC (C). In each ISC-EB-early-EC cell nests, the early EC displayed the highest expression level than the EB or the ISC (D). (E) As a control, no signal was detected by anti-Sox21a staining (red) in *Sox21a* null midgut. (F) The relative fluorescence intensity of Sox21a expression in each ISC-EB pairs and ISC-EB-Early EC cell nests (also see method). Error bars represent s.e.m. n is as indicated. ** denotes student's t test p<0.01. (G) Expression of GMR43E09>GFP (green) and Dl (red) in midgut of 5-day-old females. GFP levels were higher in EB (yellow arrowhead) than in adjacent ISC (black arrowhead). (H) A schematic model for dynamic Sox21a expression pattern: the Sox21a expression level may distinguish different state of progenitors. Sox21a expression is at the minimum in differentiation-arrested EBs which typically display a triangle shape with cell protrusions. We refer to this state of EBs as 'dormant'. Sox21a expression is elevated in EBs when EBs begin to differentiate and display oval-shaped morphology. We refer to this state of EBs as 'active'. Sox21a expression reaches its highest level in differentiating EC, and then gradually disappears as EC matures. Scale bars in **A**, **B**, 10 µm; in **E**, 20 µm.

The following figure supplement is available for figure 1:

**Figure supplement 1.** The Sox21a antibody is highly specific.

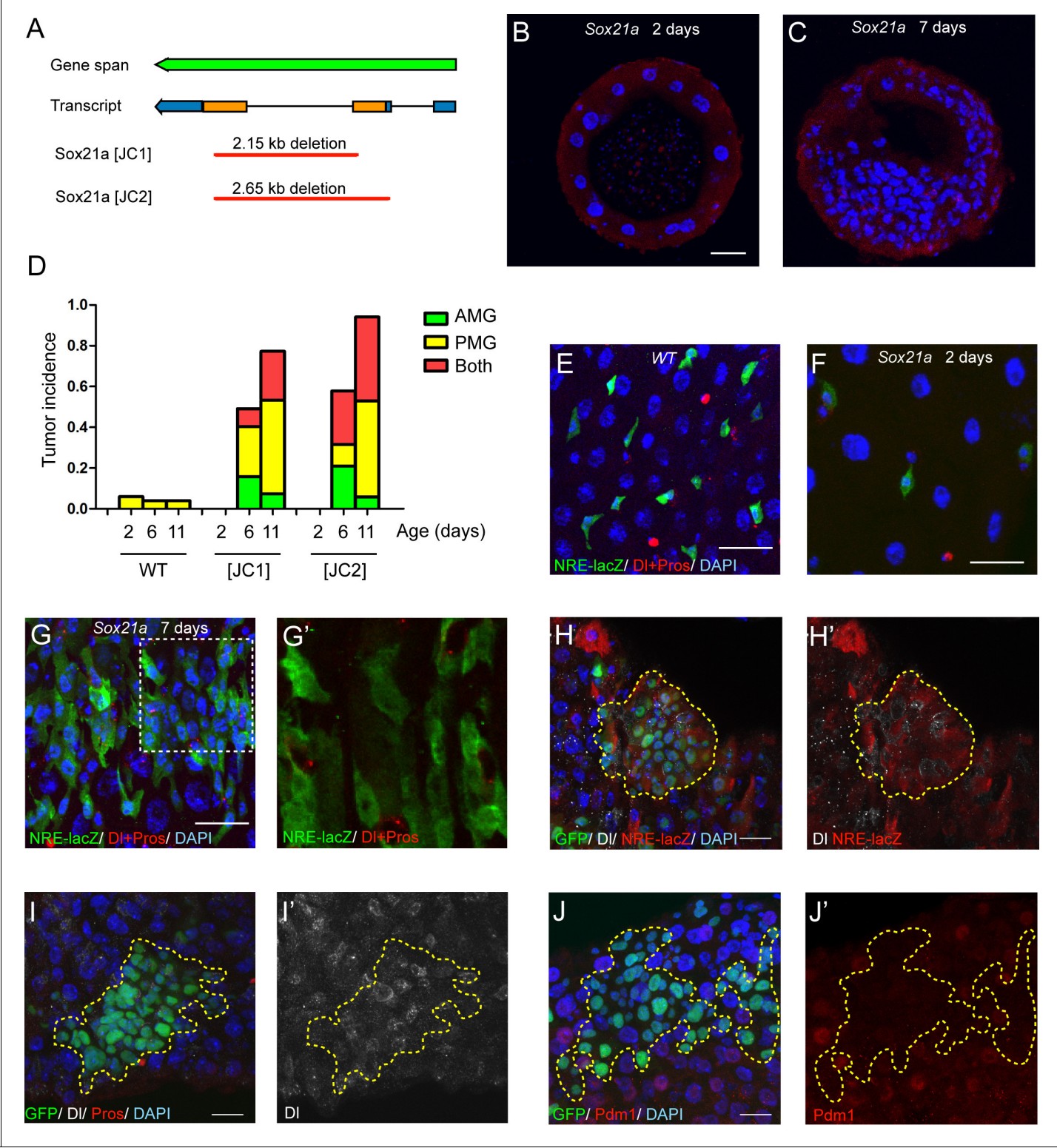

**Figure 2.** *Sox21a* mutation causes the development of intestinal tumors composed of differentiation-defective cells. (A) A diagram describing the molecular lesions of two *Sox21a* mutant alleles. Both alleles carry a large DNA fragment deletion spanning the coding region of *Sox21a*. (B–C) Cross-section of midgut from 2 day old (B) and 7 day old (C) *Sox21a* mutant flies. With age, *Sox21a* mutants developed tumors with multilayered structure (C). (D) Quantitative analysis of the tumor incidence at different regions of midgut with age. Genotypes of flies analyzed: wild type (WT), *Sox21a*$^{-/-}$ [JC1] and *Sox21a*$^{-/-}$ [JC2]. n = 40–50 guts. (E–G') Staining of NRE-lacZ (green) and Dl + Pros (red) in the midgut of WT (E), *Sox21a* mutant flies at day 2 (F) and day

*Figure 2 continued on next page*

*Figure 2 continued*

7 (**G**, **G'**) after eclosion. Compared to WT guts, *Sox21a* mutant guts did not show EB accumulation at day 2 (**E**, **F**), but showed dramatic EB accumulation at day 7 (**G**, **G'**). Accumulated lacZ⁺ cells were negative for Dl or Pros expression (**G'**). (**H**, **H'**) Staining of NRE-lacZ (red) in *Sox21a* mutant clones on day 7 after clone induction. LacZ was cell-autonomously activated. Non-cell autonomous LacZ+ cell clusters were also observed (see text). (**I**–**I'**) Staining of Dl (white), Pros (red) in *Sox21a* mutant clones. The Dl⁺ cells were scatteredly distributed in the clones, and Pros⁺ cell was rarely found within the mutant clones. (**J**, **J'**) Staining of Pdm1 (red) in *Sox21a* mutant clones.. Pdm1 expression was absent in the entire mutant clones. Scale bars: 20 μm.

intestine at day 2 after eclosion (*Figure 2E,F*). However, the NRE-lacZ⁺ cells rapidly accumulated in the epithelium of the mutant intestine at day 7 (*Figure 2G,G'*). Thus, *Sox21a* mutants postnatally develop intestinal tumors. Co-staining Dl with NRE-lacZ revealed that ISCs and EBs were intermingled with each other in EB-accumulated regions (*Figure 2G,G'*). Those observations suggest that the tumors in *Sox21a* mutant intestine are mainly composed of ISCs and EBs, and indicate that Sox21a might be required for EB differentiation. To test the differentiation capability of *Sox21a* mutant cells, we generated *Sox21a* homozygous mutant clones by mitotic recombination using the MARCM system. As expected, the mutant clones cell-autonomously gave rise to intestinal tumors which was mainly composed of NRE-lacZ⁺ cells intermingled with Dl⁺ cells (*Figure 2H,H'*). We also observed clusters of NRE-lacZ⁺ cells outside of the clones (data not shown), which could be due to a non-cell autonomous effect by the mutant clones. There was also mild increase of Dl⁺ cells that were scatteredly distributed within the mutant clones (*Figure 2I,I'*). Rarely but occasionally Pros⁺ cells could be found in the mutant clones (*Figure 2I* and data not shown), indicating that Sox21a is not absolutely required for enteroendocrine cell differentiation. Polyploid cells were also found in the clones but their size was relatively smaller than ECs (*Figure 2I–J*). Interestingly, Pdm1, an enterocyte marker (*Lee et al., 2009*), failed to be expressed in *Sox21a* mutant cells in a cell-autonomous manner (*Figure 2I,I'*). These observations suggest that Sox21a is cell-autonomously required for EB differentiation into enterocyte.

## Sox21a functions to promote EB differentiation into enterocyte

The above results demonstrate that Sox21a is required for the expression of Pdm1. Because Pdm1 expression is sufficient to induce differentiation of progenitor cell into enterocyte (*Korzelius et al., 2014*), Sox21a may function to promote enterocyte differentiation by inducing Pdm1 expression. To test this hypothesis, we generated a *UAS-Sox21a* transgene and forcibly expressed it in progenitor cells using esgGal4, an ISC- and EB-specific Gal4 driver, and examined the consequences. Tub-Gal80ᵗˢ transgene was also added to the system to allow temporal control of transgene expression. Under normal conditions, virtually all Esg>GFP⁺ cells were negative for Pdm1 expression (*Figure 3A,A'*). However, when Sox21a was forcibly expressed for 5 days, approximately 34.2 ± 1.6% of Esg>GFP⁺ cells became positive for Pdm1 expression (*Figure 3B,B',E*). The Pdm1⁺ cells were always one of the GFP⁺ ISC-EB pairs, indicating that expression of *Sox21a* could not induce Pdm1 expression in ISCs, but EBs (*Figure 3B,B'*). To test this hypothesis, we forcibly expressed *Sox21a* specifically in EBs using the Su(H)Gbe-Gal4 (NRE-Gal4) driver. Indeed, transient induction of Sox21 expression for 5 days induced approximately 72.3 ± 3.8% of NRE>GFP⁺ cells to turn on Pdm1 expression (*Figure 3D,D',E*), and these cells began to show polyploidy, indicating that they are in the process of enterocyte differentiation (*Figure 3D,D'*). Next, we generated MARCM clones in which all GFP+ cells had constitutive Sox21a overexpression. The ISC-containing clones grew normally in size and were able to produce both polyploidy enterocytes and Pros⁺ enteroendocrine cells (*Figure 3F*), suggesting that forced Sox21a expression does not inhibit ISC activity and does not block enteroendocrine cell differentiation. Taken together, these data suggest that Sox21a overexpression is not sufficient to induce differentiation of ISCs, but is able to induce precocious differentiation of EBs by inducing Pdm1 expression.

## *Sox21a* mutant EBs are non-mitotic but are tumor-initiating cells

The requirement for Sox21a in EB differentiation explains the EB accumulation phenotype in *Sox21a* mutant intestine. Rapid EB-like tumor development indicates that the mutant EBs might have re-entered the cell cycle to replicate themselves, although normally Notch-activated EBs are non-

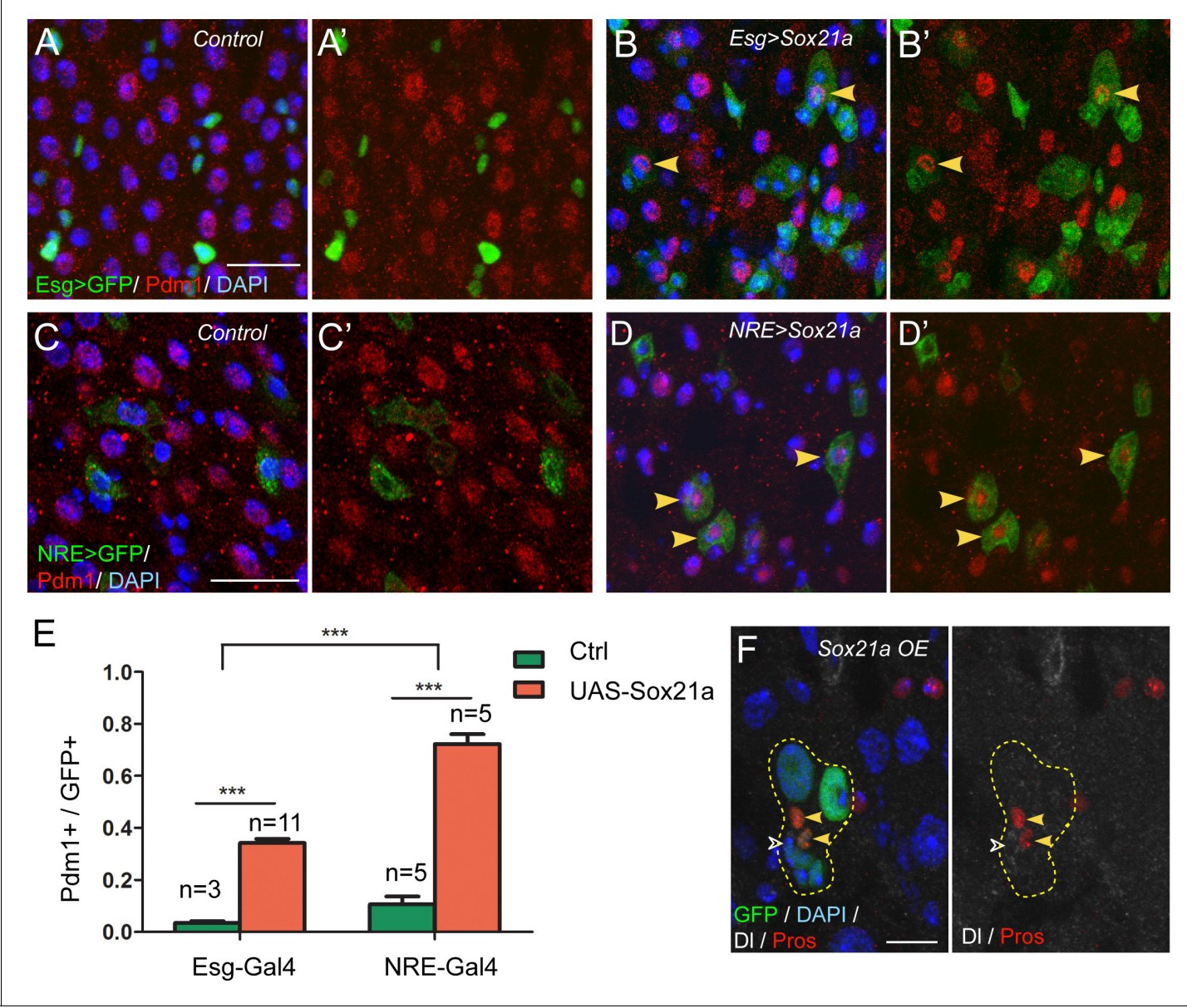

**Figure 3.** Sox21a functions to promote EB differentiation into EC. (A–B') Expression of Esg>GFP (green) and Pdm1 (red) in control (A, A') and Sox21a overexpressed guts(B, B'). Conditional expression of Sox21a in Esg+ cells for 7 days induced a part of Esg+ cells to turn on Pdm1 expression. (C–D') Expression of NRE>GFP (green) and Pdm1 (red) in control (C,C') and Sox21a overexpressed guts (D,D'). Conditional expression of Sox21a in NRE+ cells for 5 days induced most of the NRE+ cells to turn on Pdm1 expression. (E) Quantification of the percentage of Pdm1+ cells in GFP+ cells. Error bars represent s.e.m. n is as indicated. *** denotes student's t test p<0.001. (F) Staining of Dl (white) and Pros (red) in MARCM clones with Sox21a overexpression (*Sox21a OE*) on day 5 after clone induction. Dl+ ISCs and Pros+ enteroendocrine cells could be detected in Sox21a OE clones. Scale bars in (A–D), 20 μm; in F, 10 μm.

mitotic and primed for enterocyte differentiation. BrdU incorporation assay revealed that the tumor cells had strikingly high ability to incorporate BrdU (*Figure 4A,B*). As expected, the mutant intestine had exponentially increased number of mitotic cells with age (*Figure 4C*). In addition, significantly increased number of mitotic cells was found in the tumors compared to non-tumor areas (not shown). These observations indicate that the increased mitotic activity drives tumor growth. To determine the identity of mitotic cells in Sox21 mutant tumors, we co-stained the mitotic marker phospho-histone 3 (PH3) with various cell fate markers. Surprisingly, virtually all mitotic cells in the

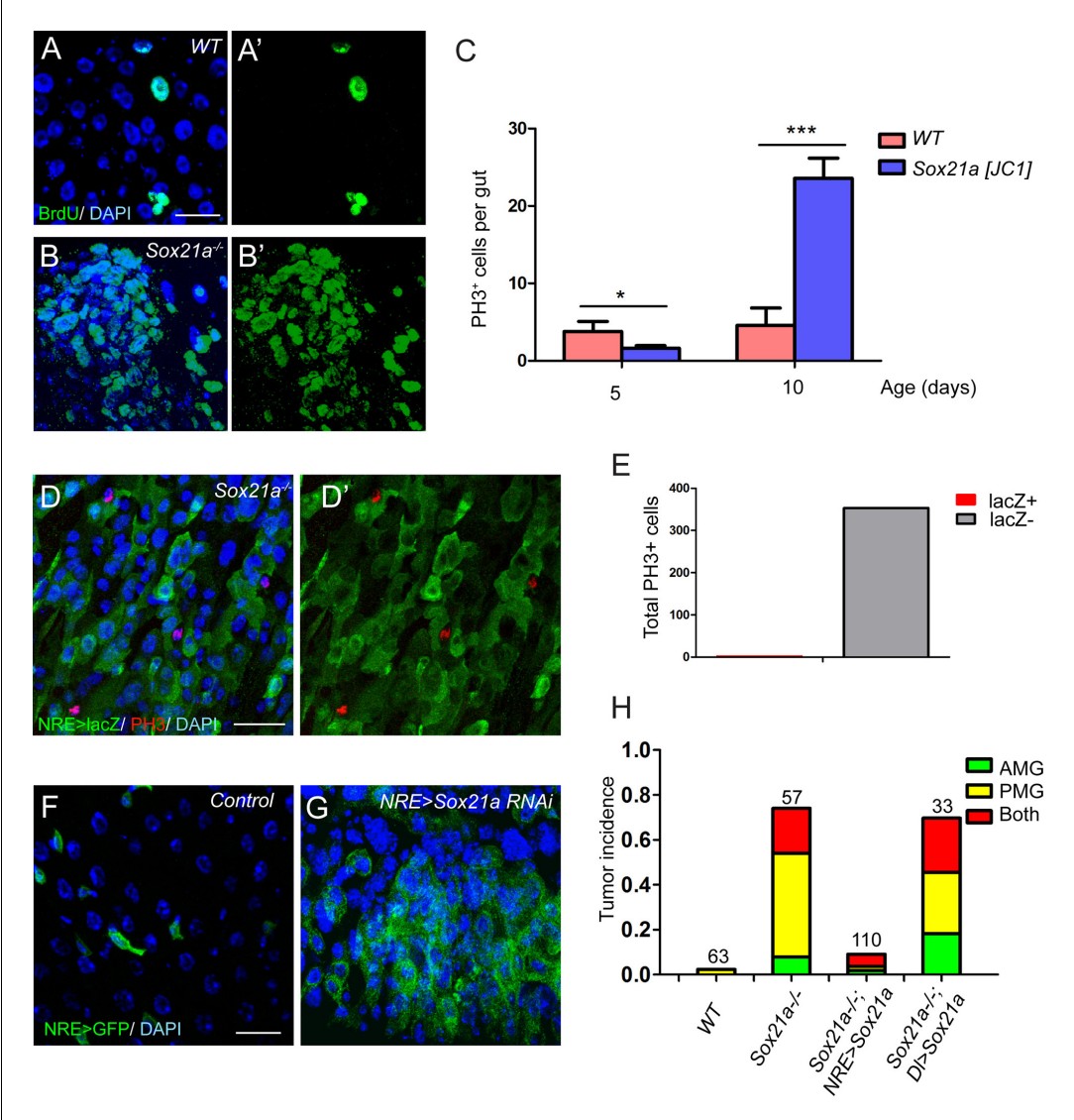

**Figure 4.** *Sox21a* mutant EBs are non-mitotic but are tumor-initiating cells. (A–B') BrdU (green) incorporation assay in WT control and *Sox21a*[-/-] gut. Compared to WT gut, *Sox21a* mutant gut showed increased BrdU incorporation in tumor regions. (C) Mitotic index of WT and *Sox21a*[-/-] midgut at day 5 and day 10 after eclosion. Error bars represent s.e.m. n = 40–50 guts. * denotes student's t test p<0.05. *** denotes student's t test p<0.001. (D–E) pH3 (red) staining in the tumor region of *Sox21a* mutant midgut (D,D'). Virtually all pH3[+] cells were NRE>GFP[-] cells (D',E). (F, G) *Sox21a*-RNAi driven by NRE-Gal4[ts] led to tumorous accumulation of GFP[+] EBs. (H) Quantitative analysis of tumor incidence in the midgut of WT control, *Sox21a*[-/-], *Sox21a*[-/-] ; *NRE>Sox21a* and *Sox21a*[-/-] ; *Dl>Sox21a* flies. Expression of *Sox21a* transgene in EB, but not in ISC, could effectively suppress tumor development in *Sox21a* mutant midgut. Total number of guts examined is as indicated. Scale bars: 20 μm.

tumor were NRE-lacZ[-] (353 out of 354) (*Figure 4D–E*). Thus, the proliferative cells that drive tumor growth are not EBs but ISCs.

One possible explanation for the increased ISC proliferation in the tumor is that, in addition to a function in EB differentiation, Sox21a also functions in ISCs to inhibit cell proliferation. However, Specifically knocking down *Sox21a* by RNAi (HMJ21395 or JF02191) in EBs using Su(H)Gbe-Gal4, UAS-GFP; Tub-Gal80[ts] (referred to as NRE-Gal4[ts]) was sufficient to induce rapid accumulation of EB-like cells in the intestinal epithelium (*Figure 4F,G*), a phenotype largely similar to that observed in *Sox21a* mutant intestine. Therefore, loss of Sox21a in EB is sufficient to induce tumor development, indicating that *Sox21a* mutant EBs are tumor-initiating cells, although they are non-mitotic cells.

If the non-mitotic EBs are indeed tumor-initiating cells in *Sox21a* mutants, we would expect to see that EB-specific transgene expression of Sox21a should be able to prevent tumor development in *Sox21a* mutant intestine. We therefore generated *Sox21a* mutant flies carrying NRE-GAL4^ts and *UAS-Sox21a* transgenes, in which the expression and therefore the function of Sox21a will be restricted in EBs in the intestinal epithelium. In another experiment, we also generated *Sox21a* mutant flies carrying Dl-GAL4^ts and *UAS-Sox21a* transgenes, in which the expression and therefore the function of Sox21a will be restricted in ISCs in the intestinal epithelium. Indeed, EB-specific expression of Sox21a largely suppressed intestinal tumor formation in both anterior and posterior midgut of *Sox21a* mutants (*Figure 4H*). By contrast, ISC-specific expression of Sox21a failed to suppress intestinal tumor formation in *Sox21a* mutants (*Figure 4H*). Taken together, these data suggest that EB is the tumor-initiating cell that is necessary and sufficient for intestinal tumor development in *Sox21a* mutant intestine.

## Loss of Sox21a in EBs causes upregulation of multiple secreted mitogens

Because *Sox21a* mutant EBs are non-mitotic yet are responsible for tumor initiation, the mutant EBs must non-cell autonomously induce ISC proliferation to promote tumor formation. To understand the underlying mechanisms, we compared the transcriptome of Esg⁺ progenitor cells with and without Sox21a depletion. Multiple secreted growth factors were found to be up-regulated by two fold or more in Sox21a-depleted cells, including Spitz (Spi), a ligand for EGFR/Ras/MAPK signaling, Upd3, a ligand for JAK/STAT signaling, and Pvf2 and Pvf1, which also induces Ras/MAPK signaling via PVR (*Figure 5A*). Interestingly, these factors all function as positive regulators of ISC proliferation during normal homeostasis and/ or in response to injury or infection (*Beebe et al., 2010*; *Biteau et al., 2011*; *Bond and Foley, 2012*; *Buchon et al., 2009*; *2010*; *Choi et al., 2008*; *Jiang et al., 2009*; *2011*; *Lin et al., 2010*; *Xu et al., 2011*). In addition, increased Spi signaling promotes tumor development from Notch mutant ISCs (*Patel et al., 2015*), and increased Upd3 signaling in EBs following the disruption of the Misshapen-Warts-Yorkie pathway also induce ISC proliferation and intestinal hyperplasia (*Li et al., 2014*). This raises an interesting hypothesis that *Sox21a* mutant EBs may induce ISC proliferation via paracrine Ras/MAPK and/or JAK/STAT signaling. In vivo validation of gene expression was subsequently conducted with available enhancer trap lines. Spi-lacZ is an enhancer trap line for Spi, and its lacZ expression was barely detectable in the wild type midgut epithelium, including ISCs and EBs, probably due to low levels of baseline expression in healthy midgut (*Figure 5B,B′*). However, lacZ expression became readily detectable in EBs following Sox21a depletion by RNAi (*Figure 5C,C′*). By contrast, we did not observe significant upregulation of Upd3-lacZ, a reporter for Upd3 expression (*Figure 5—figure supplement 1*). Interestingly, staining with phospho-ERK (pERK), a direct readout of MAPK signaling activity, revealed that the pERK signal was strongly enhanced in the diploid cells adjacent to the EB cells with Sox21a depletion (*Figure 5D,E*). Importantly, the increase of pERK activity in *Sox21a* mutant epithelium was completely suppressed when Sox21a was re-expressed in EBs (*Figure 5F,F′*), suggesting that Sox21a depletion in EB is both sufficient and necessary for enhanced pERK signaling in *Sox21a* mutant intestine. We found that conditional overexpression of Spi in EBs also produced a similar EB-like tumor phenotype, indicating that Spi upregulation could be one of the reasons responsible for the enhanced pERK signaling in ISCs (*Figure 5G,G′*). Consistent with this hypothesis, EB-specific knockdown of *spi* significantly reduced, although not completely eliminated, the ISC overproliferation phenotype in *Sox21a* mutant intestine (*Figure 5H*). As a comparison, EB-specific expression of *Sox21a* transgene produced a much stronger inhibitory effect on ISC proliferation in *Sox21a* mutant intestine, and the mitotic index was reduced to wildtype levels (*Figure 5H*). Taken together, these observations suggest that in *Sox21a* mutant intestine, loss of Sox21a in EBs causes derepression of multiple mitogens, including Spi, and others, such as Pvf2 and Pvf1, that all act as paracrine signals to induce Ras/MAPK signaling activity in ISCs and promote ISC proliferation. As a result, more mutant EBs are produced, which further propel ISC proliferation. This leads to the establishment of a positive amplification loop between ISCs and their progeny, which eventually leads to uncontrolled production of differentiation defective EBs from ISCs and consequently tumorigenesis.

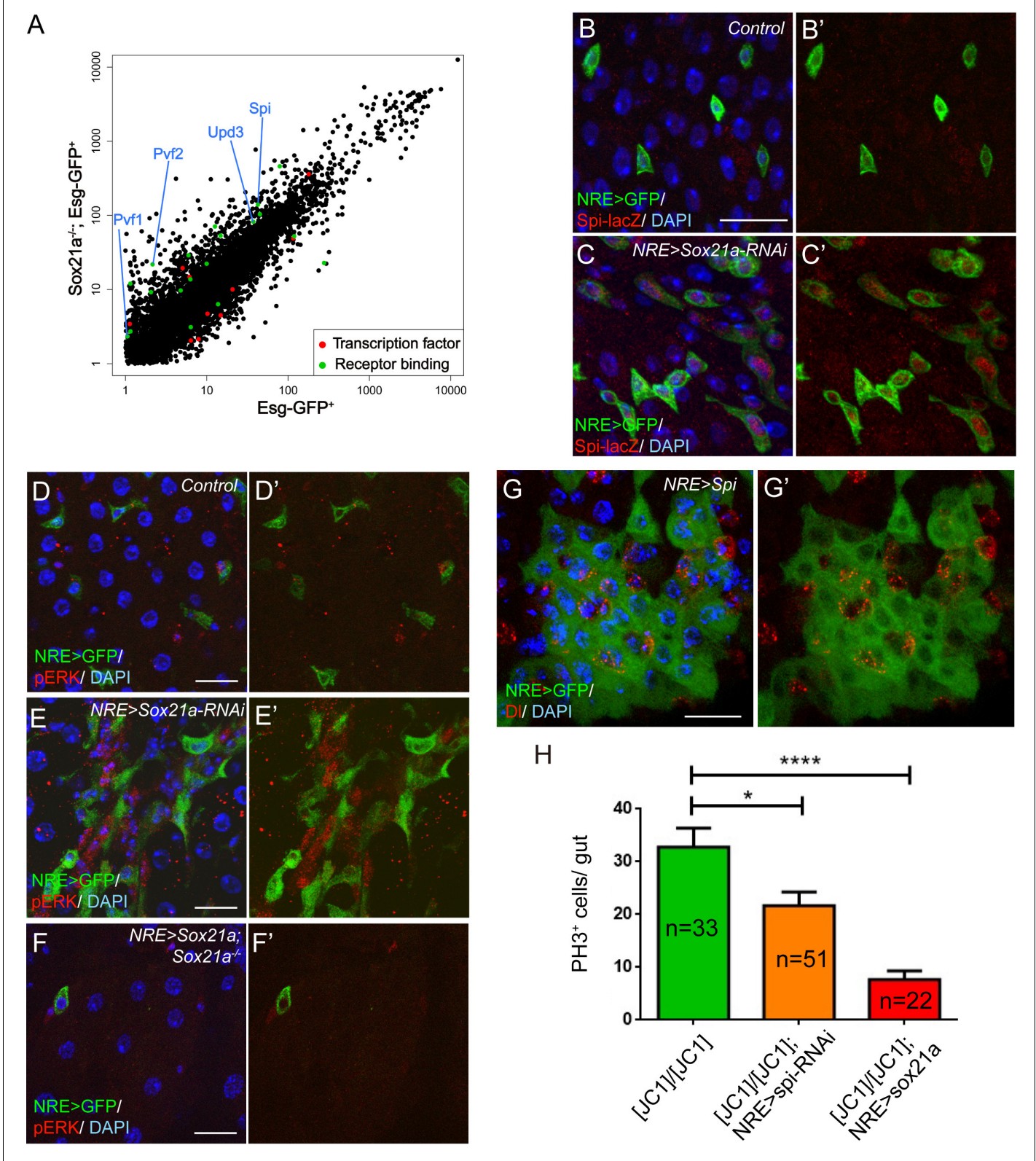

**Figure 5.** Spi/MAPK signaling is required for tumorigenesis driven by *Sox21a* mutant EBs. (**A**) A scatter plot shows the comparison of gene expression profiles of Esg-GFP+ cells in wild type and *Sox21a*-/- midgut. Green dots depict genes of receptor binding proteins with 2-fold and higher changes, and red dots depict genes of transcription factors with 2-fold and higher changes in the mutant midgut. (**B–C'**) Spi-lacZ (red) and NRE>GFP (green) expression in control and *NRE>Sox21a* RNAi midguts of 7-day-old females. Spi-lacZ expression was undetectable in the wild type midgut (**B, B'**), but

*Figure 5 continued on next page*

*Figure 5 continued*

was elevated in EBs of *NRE>Sox21a-RNAi* midgut (**C, C'**). (**D–F**) pERK (red) staining in midguts of the following genotypes: NRE>GFP (control, **D, D'**), *NRE>Sox21a* RNAi (**E, E'**) and NRE>*Sox21a; Sox21a*$^{-/-}$ (**F, F'**). EB- specific *Sox21a*-RNAi caused dramatic enhancement of pERK signal in the diploid cells adjacent to EBs, and EB-specific expression of Sox21a suppressed pERK upregulation in *Sox21a* mutant midgut. (**G, G'**) Transgenic Spi expression driven by NRE-gal4$^{ts}$ led to tumorous accumulation of EBs. Those accumulated EBs were all negative for Dl expression (**G'**). (**H**) A plot showing that either EB- specific knockdown of *spi* or EB- specific expression of *Sox21a* transgene significantly reduced ISC overproliferation in *Sox21a* mutant intestine, and *sox21a* transgene expression had a much stronger effect. Error bars represent s.e.m. n is as indicated. * denotes student's t test p<0.05. ***p<0.001. Scale bars: 20 μm.

The following figure supplements are available for figure 5:

**Figure supplement 1.** Upd3-lacZ was not significantly upregulated in *Sox21a*-depleted EBs.

**Figure supplement 2.** Sox21a was still expressed in JAK/STAT-compromised clones.

## Sox21a-Spi- mediated ISC-EB amplification loop participates in damage-induced intestinal regeneration

The regulatory role for Sox21a in Spi signaling indicates that Sox21a not only functions to promote EB differentiation, but may also provide a feedback mechanism from differentiating EBs for controlling ISC activity, and thereby intestinal homeostasis. We therefore tested whether the expression of Sox21a could be regulated upon tissue damage. Feeding flies with Dextran sulfate sodium (DSS) has been established as an effective mean to induce intestinal damage and inflammation, which then triggers ISC activation for accelerated epithelial repair (*Amcheslavsky et al., 2009*). A relatively rapid damage and repair process was set for this experiment (*Figure 6A*). Interestingly, following DSS treatment, Sox21a expression was rapidly downregulated in EBs (*Figure 6B,B',C,C',I*). This downregulation was accompanied by massive EB accumulation in the epithelium resulted from increased ISC proliferation (*Figure 6C,C'*). However, during the recovery phase following the DSS treatment, Sox21a was dramatically upregulated in the accumulated EBs (*Figure 6D,D',I*), and differentiating ECs (*Figure 6E,E'*). Interestingly, spi-lacZ showed exactly the reciprocal expression pattern to Sox21a during the damage and regeneration process, as its expression was activated during the DSS treatment, and diminished at the recovery phase (*Figure 6F–H"*). We also examined the expression of the Sox21a expression reporter GMR43E09>GFP during the damage and repair process. Consistent with the changes in Sox21a expression, GMR43E09>GFP was transiently downregulated right after the damage and then upregulated 1.5–2 days after the damage (*Figure 6—figure supplement 1*).

These observations indicate that the Sox21a-Spi regulatory pathway could mediate damage-induced intestinal regeneration. To functionally test this hypothesis, we asked whether reversing the expression of Sox21a or Spi in EBs could affect intestinal regeneration following damage. Specific depletion of Spi in EBs significantly reduced the proliferative response of ISCs following DSS treatment (*Figure 6J*). On the other hand, overexpression of Sox21a in EBs almost completely eliminated the proliferative response of ISCs (*Figure 6J*). These results suggest that dynamic regulation of the Sox21-Spi pathway is critical for effective intestinal regeneration after damage. Compared to the effect by *EB>Sox21a*, incomplete suppression of ISC proliferation by *EB>Spi-RNAi* may indicate that Spi is merely one of the factors downstream of Sox21a that mediate damage-induced ISC proliferation.

Taken together, our data suggest a model in which an ISC-EB feedback amplification loop controlled by dynamic expression of Sox21a regulates intestinal homeostasis, regeneration and tumorigenesis (*Figure 7*). Normally, Sox21a is expressed in differentiating EBs to promote enterocyte differentiation. Following tissue damage, Sox21a is temporally downregulated in EBs to allow the production of mitogens, which then act back on ISCs to promote ISC proliferation. This leads transient activation of ISC-EB amplification loop for accelerated production of progenitor cells prepared for epithelial differentiation. At the recovery phase, Sox21a is upregulated to shut down the ISC-EB amplification loop and promote cell differentiation and homeostasis re-establishment. In *Sox21a* mutant intestine, the ISC-EB feedback amplification loop is continuously active, which drives massive production of differentiation-defective cells and tumorigenesis (*Figure 7*).

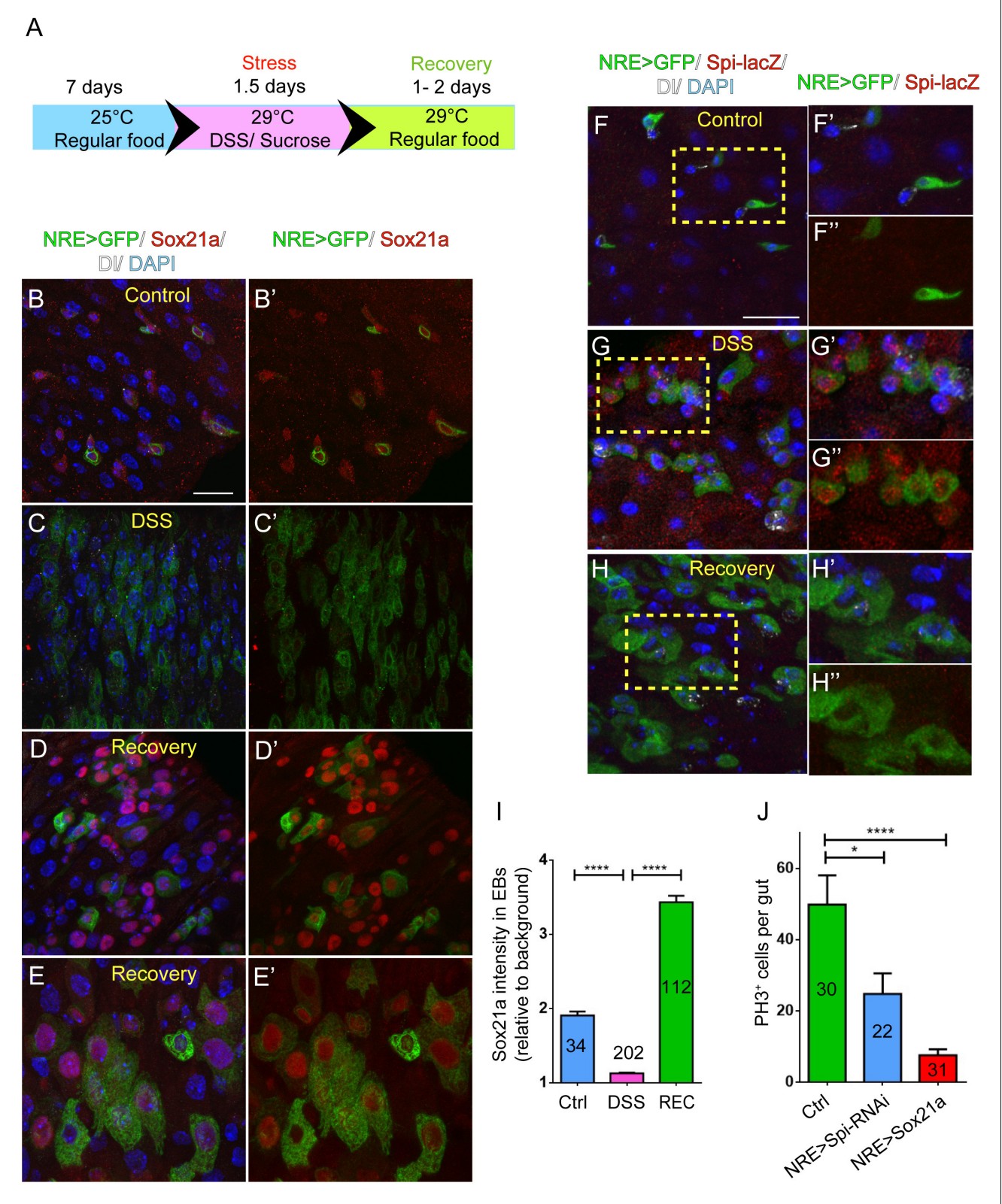

**Figure 6.** Sox21a-Spi- mediated ISC-EB amplification loop participates in damage-induced intestinal regeneration. (**A**) The scheme of damage induction and recovery. (**B**) (**B–E'**) Sox21a (red) expression in the midgut of flies fed with sucrose-soaked diet (**B**, **B'**) and DSS-soaked diet (**C**, **C'**) as well as the flies of 2 day recovery after DSS feeding (**D**, **D'**, **E**, **E'**). Compared with sucrose-treated control gut (**B**, **B'**), DSS treatment showed rapid decline of Sox21a expression in EBs (**C**, **C'**). During the recovery phase, Sox21a was dramatically upregulated in EBs (**D**, **D'**) and differentiating ECs (**E**, **E'**). (**F–H"**)
*Figure 6 continued on next page*

*Figure 6 continued*

Spi-lacZ (red) expression in the midgut of flies treated with sucrose (**F–F"**) or DSS (**G–G"**) as well as the flies at the recovery phase after DSS treatment (**H–H"**). Compared with sucrose-treated midgut, in which Spi-lacZ expression was undetectable (**F–F"**), DSS treatment induced Spi-lacZ expression (**G–G"**). Spi-lacZ expression was shut down again at the recovery phase (**H–H"**). (**I**) Fluorescence intensity of Sox21a expression in EBs relative to background in the midgut of flies fed with sucrose, DSS and flies at the recovery phase after DSS treatment. Sox21a expression in EB was virtually reduced to background levels in DSS-induced damage phase, and then massively upregulated during the recovery phase (REC). Error bars represent s. e.m. n is as indicated. **** denotes student's t test $p<0.0001$ (**J**) Quantification of pH3$^+$ cells in midguts of indicated genotypes. EB-specific depletion of Spi could partially reduce DSS-induced mitosis, while EB-specific transgene expression of Sox21a could strongly inhibit DSS-induced mitosis. Error bars represent s.e.m. n is as indicated. * denotes student's t test $p<0.05$. ***$p<0.001$. Scale bars: 20 μm.

The following figure supplement is available for figure 6:

**Figure supplement 1.** GMR43E09>GFP (green) expression in the midgut of flies treated with sucrose (**A–A'**) or DSS (**B, B'**) as well as flies at the recovery phase after DSS treatment (**C–C'**).

## Discussion

Sox family proteins in metazoan are divided into different groups based on their similarity in biochemical properties, and Sox21a belongs to the SoxB2 subgroup whose function is relatively less studied compared to the most related SoxB1 subgroup proteins, such as the pluripotency factor Sox2 (*McKimmie et al., 2005*; *Sarkar and Hochedlinger, 2013*). By cellular and genetic analysis, here we have characterized the functions of Sox21a, a member of the SoxB2 group in *Drosophila* and revealed two major roles. First, Sox21a is essential for EC differentiation from EB. Sox21a protein is mainly expressed in differentiating EBs, a pattern that is consistent with its role in EB differentiation. Loss of Sox21a causes accumulation of undifferentiated cells that fail to express Pdm1, the EC marker. The majority of these mutant cells remain as diploid EBs and some begin to show polyploidy, indicating that mechanisms controlling EC differentiation and cell ploidy can be uncoupled. On the other hand, forced transgene expression of Sox21a in EBs accelerates their differentiation into ECs as evidenced by precocious expression of Pdm1. However, forced expression of Sox21a in ISCs does not induce their differentiation, suggesting that Sox21a is necessary but not sufficient for inducing EC differentiation from ISCs. One possible explanation for this is that EC differentiation requires both Sox21a and Notch activity. Indeed, although Notch activity is known to be both necessary and sufficient for inducing EC differentiation from ISCs, Notch activated EBs that have already been presented along the length of midgut in young flies remain dormant for several days until there is a need for cell replacement (*Antonello et al., 2015*; *Hochmuth et al., 2011*). These observations indicate that activation of Notch alone is not sufficient to induce EB differentiation, but only primes EB for EC differentiation. Notch-activated EBs will stay in undifferentiated state until Sox21a is activated, which then promotes differentiation of the primed EBs into ECs.

Another surprising role for Sox21a revealed in this study is that it provides a feedback regulation of ISC activity by suppressing mitogenic signals from differentiating EBs. This function is important for controlling the strength of the ISC-EB amplification loop for balanced self-renewal of ISCs and differentiation of EBs. Because Sox21a is also required for EB differentiation, disruption of this function will cause sustained activation of the ISC-EB amplification loop as well as blocked EB differentiation, leading to the formation of EB-like tumors. Importantly, following epithelial damage, Sox21a is quickly downregulated in EBs. This allows temporal activation of the ISC-EB loop for rapid production of progenitor cells prepared for epithelial repair. During recovery, Sox21a is then temporally upregulated in EBs. This not only stops the ISC-EB amplification loop to avoid excessive EB production, but also accelerates cell differentiation for epithelial repair. Therefore, Sox21a does not simply act as a tumor suppressor in intestine. It is dynamically regulated to control the process of epithelial regeneration in response to various environmental changes via regulating the strength of the ISC-EB amplification loop. Because Sox21a expression in intestine is dynamic during normal adult development, it is conceivable that its expression could possibly be influenced by physiological changes, such as food intake and activity of symbiotic bacteria, and fine-turning Sox21a activity could be important for maintaining regular epithelial turnover. How Sox21a expression is regulated is unclear, but signaling pathways that are implicated in regulating intestinal regeneration, such as JAK/STAT or EGFR/Ras pathways are potential candidate regulators, especially JAK/STAT, which is known to

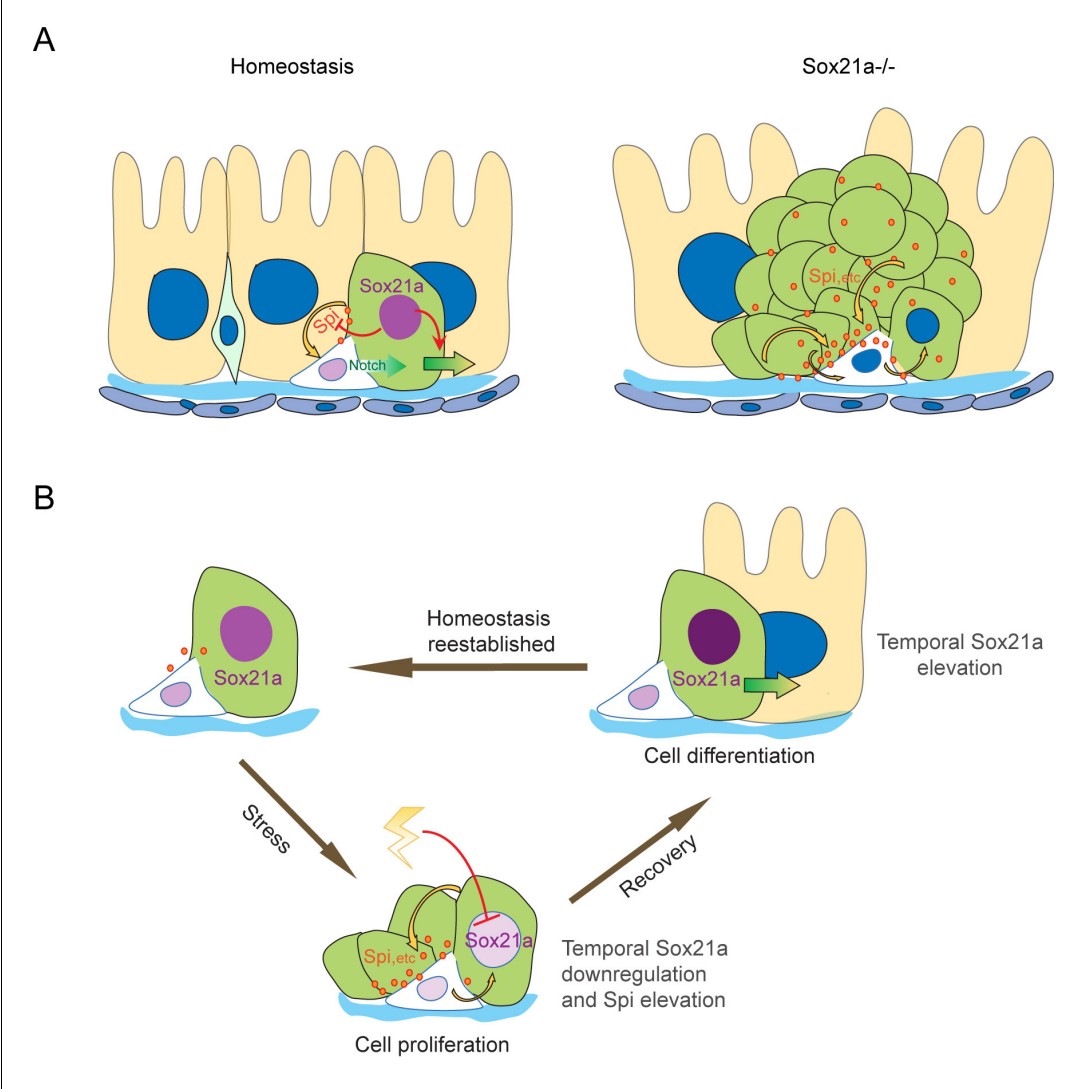

**Figure 7.** Schematic models of the ISC-EB feedback loop during normal homeostasis, regeneration and tumorigenesis. (**A**) During normal homeostasis (left panel), ISC and EB typically exist as a progenitor pair. Sox21a is expressed in EB to promote EB differentiation and at the same time to restrict ISC activity by inhibiting paracrine signals, such as Spi, etc.. In *Sox21a* mutant midgut (right panel), continuous paracrine signals drives continuous activation of the ISC-EB loop, leading to massive production of differentiation-defective cells and tumorigenesis. (**B**) Sox21a-mediated amplification loop is employed in damage-induced intestinal regeneration. Tissue damage causes temporal downregulation of Sox21a in EB and consequently derepression of mitogenic factors, which act in a paracrine manner to promote ISC proliferation. This leads to a temporal activation of the ISC-EB amplification loop for rapid production of progenitor cells prepared for epithelial differentiation. During the recovery after the damage is withdrawn, Sox21a is temporally upregulated in EBs to promote epithelial differentiation and homeostasis reestablishment.

be essential for EB differentiation. During the preparation of this manuscript, two groups have reported the function of Sox21a in *Drosophila* midgut (*Meng and Biteau, 2015*; *Zhai et al., 2015*). Meng and Biteau did not observe the tumor suppressive function of Sox21a, possibly because they used a weak mutant allele of *Sox21a* in their study. Notably, Zhai et al., observed a similar tumor suppressive role for Sox21a as reported here. Interestingly, they suggest that Sox21a is regulated by JAK/STAT signaling, as Sox21a transgene expression is able to rescue the differentiation defects caused by disrupted JAK/STAT signaling. However, we found that Sox21a was still expressed in JAK/STAT compromised EBs (*Figure 5—figure supplement 2*). Therefore, how Sox21a is regulated during normal homeostasis and regeneration remains to be further explored, and it is possible that Sox21a could be controlled by a combination of regulators in a cell type-specific manner, with

different mechanisms in ISCs and EBs. Because Sox proteins commonly function together with other cell type-specific co-factors in regulating gene transcription (*Kamachi and Kondoh, 2013*; *Sarkar and Hochedlinger, 2013*), Sox21a could function with different co-factors in ISCs and EBs. These are interesting questions worthy of further investigation.

Cells in a given tumor are usually heterogeneous and based on the ability to initiate tumors, tumor cells can be divided into tumor-initiating cells and non-tumor-initiating cells. Here in this case, Sox21 mutant EBs can be regarded as the tumor-initiating cells in vivo. Depleting Sox21a specifically in EBs is sufficient to initiate EB-like tumors. Conversely, restoring Sox21a function specifically in EBs is sufficient to prevent tumor development in *Sox21a* mutant intestine. However, unlike typical tumor-initiating cells, *Sox21a* mutant EBs are post-mitotic cells. In addition, their ability to initiate tumors depends on the activity of local ISCs. Therefore, this study also reveals a novel example of tumor-initiating cells in vivo that do not divide themselves, but can "propagate" themselves by utilizing local stem cells.

In short, by studying the function of Sox21a in *Drosophila* ISC lineages, we identified a novel feedback amplification loop between stem cells and their progeny that mediates epithelial regeneration and tumorigenesis. It has long been suggested that tissue regeneration and tumorigenesis are intimately associated, although the mechanistic connection is still obscure (*Oviedo and Beane, 2009*). The Sox21a-Spi mediated- ISC-EB amplification loop revealed here may provide a simple example of potential mechanisms that could connect tissue regeneration with tumorigenesis: transient activation of the stem cell- progeny amplification loop promotes regeneration, whereas sustained or irreversible activation of the amplification loop promotes tumorigenesis. We propose that this could be a general mechanism underlying tissue regeneration and tumorigenesis in other tissues, including that in mammals and humans.

## Materials and methods

### Fly stocks

The following stocks were used in this study: UAS-Sox21a-RNAi (BDSC, #53991, #31902) (*Ni et al., 2011*); esg-Gal4,UAS-GFP (gift from Shigeo Hayashi); Su(H)-Gbe-lacZ (gift from Sarah Bray); Su(H)-Gbe-Gal4 and Dl-Gal4 (gift from Xiankun Zeng and Steven Hou) (*Zeng et al., 2010*); GMR43E09-Gal4 (BDSC, #46247); dome^G0468 FRT19A (*Lin et al., 2010*); UAS-sSpi (gift from Talila Volk); UAS-Spi-RNAi (VDRC, #v3922); Spi-lacZ (gift from Henry Sun). Esg-GFP (GFP trap, gift from Lynn Cooley). These stocks were generated in this study: Sox21a^JC1; Sox21a^JC2; FRT2A Sox21a^JC1; UAS-Sox21a; Upd3-lacZ was generated as previously described (*Jiang et al., 2009*).

### Generation of *Sox21a* mutant and transgenic lines

Two *Sox21a* mutants were generated by Cas9 mediated gene knock out described before (*Kondo and Ueda, 2013*). The following guide RNAs were used:

For *Sox21a*^JC1:

5'-GGAGGCGCGCCTGTAGGTCC-3' and 5'- GAATGGACGCTTCTGTCCTT-3'

For *Sox21a*^JC2:

5'-GGAGGCGCGCCTGTAGGTCC-3' and 5'- GATGCCGGGCGCGGAGTCAA-3'

To construct the *Sox21a* transgenic flies, the following primers were used to amplify the coding region of *Sox21a* from cDNA:

5'- ACGGTGAATCGGCAGTCTAA-3' and 5'- AGCCATTTGTTTGGGTTCCAG-3'

The 2.2 kb PCR product was cloned into pUAST vector. Transgenic fly strains were generated by P element-mediated germline transformation.

### Preparation of rabbit polyclonal anti-Sox21a

Polyclonal antibody against Sox21a was generated from rabbit using the following synthetic peptide: CHPHHVQLAAATLSAKYGFGS. The cysteine residual that was added at the N terminal end of the peptide was used to conjugate keyhole limpet hemocyanin (KLH). Serum obtained from immunized rabbit was purified by antigen affinity chromatography. Purified anti-serum at final dilution of 1:1000 was used.

## Mosaic analysis

The binary GAL4/UAS system was used for spatial and temporal control of transgene expression in intestine (*Brand and Perrimon, 1993*; *McGuire et al., 2003*). MARCM system was used to generate mitotic clones in intestinal epithelium (*Lee and Luo, 1999*). MARCM clones were induced in 2–3 day old female progenies by 37°C heat shock for 1 hr. For the binary expression system, crosses were made at 18°C. 2–3 day old female progenies with desired genotype were then transferred to 29°C, cultured with regular corn meal with yeast paste and transferred every two days prior to dissection and analysis.

## Immunostaining

Immunostaining of *Drosophila* midgut was performed as previously described (*Lin et al., 2008*). The following primary antibodies were used in this study: mouse anti-Dl (DSHB, 1:100 dilution); mouse anti-Pros (DSHB, 1:300); mouse anti-phospho-Histone H3 antibody (Cell Signaling Technology, 1:500); rabbit anti-pERK(Cell Signaling Technology, 1:200); rabbit polyclonal anti-lacZ antibody (Cappel,1:6000); rabbit anti-Pdm1 (gift from Xiaohang Yang, 1:1000); rabbit anti-Pros (gift from Yuh-Nung Jan, 1:1000). Secondary antibodies used in this study: goat anti-rabbit or anti-mouse IgGs-conjugated to Alexa (568 or Cy5) (Molecular Probes, 1:300). Images were captured using a Zeiss LSM510 confocal microscope. All images were adjusted in Adobe Photoshop and assembled in Illustrator.

## Fluorescence intensity measurement

The Image J software (National Institutes of Health, Bethesda, MD, USA, http://rsb.info.nih.gov/ij/) was used to measure the fluorescence intensity of Sox21a staining. The relative fluorescence intensity was calculated as the corrected fluorescence intensity of EB or Early EC divided by the corrected fluorescence intensity of ISC in the same cell nests. The corrected fluorescence intensity is the average of fluorescence intensity of each cell's nuclear region minus the average of background fluorescence intensity.

## BrdU Labeling

2 day old female flies were cultured on standard corn meal supplemented with 2 mg/mL BrdU (Sigma-Aldrich) for 5 days, transferred every day. Female guts were dissected and fixed in 4% formaldehyde for 30 min and followed by DNase I treatment for 15 min at 37°C. The samples were then washed 3 times with PBT and incubated with a rat anti-BrdU (Abcam) overnight at 4°C.

## DSS feeding

7 days old female flies were collected and starved for 10 hr and then were cultured with 5% sucrose solution with or without 5% DSS (MP Biomedicals) soaked in kimwipe paper at 29°C for one and half days. Half of the flies were then dissected for analysis (damage phase), and another half were continuously cultured on regular corn meal supplied with yeast paste for one to two days at 29°C before dissection and analysis (recovery phase).

## FACS and RNA-seq

Profiling of intestinal progenitor cells were performed according to the method described previously (*Dutta et al., 2013*). The lines of Esg-GFP and Esg-GFP; *Sox21a*$^{-/-}$ were used to harvest intestinal progenitor cells. Hundreds of guts were dissected in DEPC-PBS on ice within 1 hr and digested with 1 mg ml$^{-1}$ Elastase (Sigma, cat. no. E0258) in 1 hr at 25°C. Dissociated cells were pelleted at 400$g$ for 20 min, resuspended in DEPC-PBS, filtered with 70 µm filters (BD Falcon) and sorted using a FACS Aria II sorter (BD Biosciences). Esg-GFP fusion was used to express green fluorescent protein (GFP) to sort each cell population, using $w^{1118}$ midgut to set fluorescence gate. For each of the three biological replicates, about 250,000 sorted cells were used to harvest total RNA with the Direct-zol™ RNA MiniPrep kit (Zymo research, cat. no. E0258), as described in detail at Bio-protocol (*Chen et al., 2016*). Illumina TruSeq RNA Sample Prep Kit (Cat#FC-122-1001) was used with 50 ng of total RNA for the construction of sequencing libraries.

## RNA-seq data analysis

Single-end reads were mapped to the Drosophila melanogaster genome (Release 5) using TopHat (v2.0.10). The GTF annotations from the Ensembl release (BDGP5) were also supplied to tophat using the -G flag to allow Tophat to utilize known splice junctions. Each sequencing experiment generated an average 11.18 million raw reads, and 86% was successfully mapped for each experiment. Gene expression was quantified by the number of reads that fall into the exons. The results are normalized to RPKM (reads per kilobase of exon model per million mapped reads) using Cufflinks (v2.2.1). Differentially expressed genes were identified by using the following criteria: P value <= 0.05, Fold change >= 2.

## Acknowledgements

We thank the members of fly community as cited in Materials and methods for generously providing fly stocks and/ or antibodies, the Bloomington *Drosophila* Stock Center, Tsinghua Fly Center, Vienna *Drosophila* RNAi Center (VDRC), and Developmental Studies Hybridoma Bank (DSHB) for reagents, NIBS antibody center for generating the Sox21a antisera, and members of the Xi laboratory for helpful comments. This work was supported by China Ministry of Science and Technology (National Basic Science 973 grants 2011CB812700 and 2014CB850002 to R.X.) and National Natural Science Foundation of China (31501105 to N.X.).

## Additional information

### Funding

| Funder | Grant reference number | Author |
|---|---|---|
| National Natural Science Foundation of China | 31501105 | Na Xu |
| Ministry of Science and Technology of the People's Republic of China | 973 projects 2014CB850002 | Rongwen Xi |
| Ministry of Science and Technology of the People's Republic of China | 973 projects 2011CB812700 | Rongwen Xi |

The funders had no role in study design, data collection and interpretation, or the decision to submit the work for publication.

### Author contributions

JC, Designed the research, Performed experiments, Wrote the paper, Conception and design, Analysis and interpretation of data, Drafting or revising the article; NX, Performed experiments, Acquisition of data, Analysis and interpretation of data; HH, Analyzed the high-throughput sequencing data, Acquisition of data, Analysis and interpretation of data; TC, Analyzed the high-throughput sequencing data, Analysis and interpretation of data; RX, Designed the research, Wrote the paper, Conception and design, Analysis and interpretation of data, Drafting or revising the article

### Author ORCIDs

Rongwen Xi, http://orcid.org/0000-0001-5543-1236

## Additional files

### Major datasets

The following dataset was generated:

| Author(s) | Year | Dataset title | Dataset URL | Database, license, and accessibility information |
|---|---|---|---|---|
| Jun Chen, Rongwen | 2016 | Transcriptome profiling of | http://www.ncbi.nlm.nih. | Publicly available at |

| Xi | Drosophila Wild Type and Sox21a-/- intestinal progenitor cells | gov/geo/query/acc.cgi? acc=GSE76621 | the NCBI Gene Expression Omnibus (Accession no: GSE76621). |
|---|---|---|---|

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
