## [Decision Letter]

Thank you for submitting your work entitled "A feedback amplification loop between stem cells and their progeny promotes tissue regeneration and tumorigenesis" for consideration by *eLife*. Your article has been favorably evaluated by Sean Morrison as the Senior editor and three reviewers, including Bruce Edgar and Allan Spradling, who is a member of our Board of Reviewing Editors.

The reviewers have discussed the reviews with one another and the Reviewing Editor has drafted this decision to help you prepare a revised submission.

Summary:

The authors describe a function for the *Sox21a* gene in EB differentiation and in tissue stress response in the *Drosophila* midgut. When Sox21a is disrupted, EB differentiation is blocked and EBs accumulate and produce factors including Spitz that stimulate the proliferation of nearby ISCs and lead to intestinal tumors. Interestingly, the authors propose that this sequence of events is homeostatic when the gut is stressed. Following dextran sulfate feeding, Sox21a expression drops and EBs accumulate, presumably via the same mechanism, and stimulate ISC division in a manner that can be diminished by reducing Spi.

Essential revisions:

Recently, Zhai et al. (2015) Nature Comm. 10219 published a similar description of sox21-induced tumors and Meng and Biteau (2015) Cell Rep. 13, 906 characterized the role of Sox21a in response to midgut damage. Many of the experiments and observations reported here were previously described in these papers. Furthermore, Chen et al.'s proposal that transient tumor production might be functional is not fully consistent with these studies. In particular, Meng and Biteau reported that Sox21a is upregulated within 24hr following midgut stress, and consistent with this, Zhai et al. showed that JAK-STAT signaling, which is typically induced by midgut stress, induces Sox21a expression. Furthermore, Upd2 was shown to be the primary paracrine stimulator of ISC proliferation, with Spi (and Upd3) having only a relatively small functional effect.

Consequently, it is critically important in a revised version to resolve whether the "ISC-EB feedback amplification loop" is "dynamically regulated during normal tissue homeostasis" and "critical for damage-induced intestinal regeneration" as the authors propose. Two experiments would make this possible:

1) A critical step toward proving that Sox21a participates in a regenerative response is resolving whether Sox21a expression declines (as proposed here) or increases during normal development and after stress/injury (as suggested by the above papers). This can be done be expanding the quantification experiments in Figure 1. To first assess Sox21a expression during normal adult differentiation, show images as in Figure 1, but of cell nests of 3 or 4 cells (instead of just doublets as currently) with 1 Delta + cell, to see if the staining of Sox21a protein is not detectable in the nearest neighbor EB but appears in the older, bigger differentiating EBs on the way to become ECs. The data on the relative levels of Sox21a staining in the successive EBs should be quantitated in a graph with a reasonable N value.

The Sox21a stainings in the current Figure 1 are too faint. In this new Figure 1, provide better images, and show separate tracks of Sox21a alone, where the staining can be clearly seen and the specificity of staining evaluated (i.e. show adjacent cells and include a *Sox21a* mutant control).

The same experiment should then be performed, but after labeling has started and 2-3 EBs generated and Sox21a expression has turned on, DSS treatment should be initiated. The pattern of Sox21a expression 1-2 days later should then be determined and quantitated as above. The authors' model predicts that damage induction will cause a transient decline in Sox21a staining not seen in the control, as the proposed ISC-EB loop is set up and starts to function.

2) The second experiment would be to more clearly document a role for Spi in Sox21a-mediated ISC division. In Zhai's paper, they found that knockdown of Upd2 in the *Sox21a* mutant background significantly repressed the ISC proliferation, whereas Spi knockdown had only a small effect. The importance of Spi in the ISC-EB feedback loop should be measured quantitatively.

---

## [Author Response]

Essential revisions:

*Recently, Zhai et al. (2015) Nature Comm. 10219 published a similar description of sox21-induced tumors and Meng and Biteau (2015) Cell Rep. 13, 906 characterized the role of Sox21a in response to midgut damage. Many of the experiments and observations reported here were previously described in these papers. Furthermore, Chen et al.'s proposal that transient tumor production might be functional is not fully consistent with these studies. In particular, Meng and Biteau reported that Sox21a is upregulated within 24hr following midgut stress, and consistent with this, Zhai et al. showed that JAK-STAT signaling, which is typically induced by midgut stress, induces Sox21a expression. Furthermore, Upd2 was shown to be the primary paracrine stimulator of ISC proliferation, with Spi (and Upd3) having only a relatively small functional effect.*

By studying the suppressive function of Sox21a, we have revealed a novel feedback amplification loop mediated by Sox21a-Spi, a novel pathway that is dynamically regulated to control intestinal homeostasis and damage repair, whose dysregulation leads to tumorigenesis. Zhai et al. observed a similar tumor suppressive function of Sox21a, and described how tumor cells communicate with ISCs to propel tumor growth. However, they did not identify the ISC-EB feedback amplification loop that is dynamically regulated to promote intestinal regeneration. Zhai et al. suggested Upd2 as the primary paracrine signal for ISC division. However, we did not observe upregulation of Upd2 in Sox21a mutant EBs and neither did they. Therefore, the effect by Upd2 RNAi observed by Zhai et al. could be due to Sox21a-independent mechanisms (this will be further discussed later). Meng and Biteau did not observe the tumor suppressive function of Sox21a, possibly because they used a weak mutant allele of Sox21a in their study. They did analyze the Sox21a expression pattern, but not in explicit detail, which is rather vital in order to reveal its dynamic nature and its participation in homeostatic control. We analyzed Sox21a at various ages and revealed its dynamic expression during adult development. We examined its expression at different time points during the process of damage-induced regeneration, and discovered its downregulation in the damage phase and upregulation in the recovery phase. As described later, we used a new DSS-treatment protocol, and similar to Meng and Biteau’s observation, we found that Sox21a was upregulated as early as 1.5 days following damage. Taken together, although there are some similar results described before, we believe that the major mechanisms reported here are still novel and significant, and may have a significant impact on our understanding of stem cells in tissue regeneration and tumorigenesis.

Consequently, it is critically important in a revised version to resolve whether the "ISC-EB feedback amplification loop" is "dynamically regulated during normal tissue homeostasis" and "critical for damage-induced intestinal regeneration" as the authors propose. Two experiments would make this possible:

1) A critical step toward proving that Sox21a participates in a regenerative response is resolving whether Sox21a expression declines (as proposed here) or increases during normal development and after stress/injury (as suggested by the above papers). This can be done be expanding the quantification experiments in Figure 1. To first assess Sox21a expression during normal adult differentiation, show images as in Figure 1, but of cell nests of 3 or 4 cells (instead of just doublets as currently) with 1 Delta + cell, to see if the staining of Sox21a protein is not detectable in the nearest neighbor EB but appears in the older, bigger differentiating EBs on the way to become ECs. The data on the relative levels of Sox21a staining in the successive EBs should be quantitated in a graph with a reasonable N value.

We have expanded the quantification experiments in Figure 1. Sox21a expression in the midgut was highly dynamic during adult development. As shown previously, its expression was largely undetectable in the midgut of newly eclosed flies and young flies of two to three days old. Its expression began to appear with age and at 4-5 days old, weak Sox21a expression appeared specifically in Dl+ ISCs and Notch-activated EBs (Figure 1), and with age, the EB began show a higher expression level than the ISC in the same ISC-EB pair (Figure 1). At 7 days or older, cell nests of 3 or 4 cells, in which the early still retained NRE>GFP expression, began to appear. In these cell nests, the early EC usually displayed the highest Sox21a expression level than the EB or the ISC in the same ISC-EB-early-EC cell nests (Figure 1). The relative florescence intensity measurements for ISC, EB and early ECs in the ISC-EB pair or 3-cell nests were shown in Figure 1, and differences are statistically significant.

*The Sox21a stainings in the current Figure 1 are too faint. In this new Figure 1, provide better images, and show separate tracks of Sox21a alone, where the staining can be clearly seen and the specificity of staining evaluated (i.e. show adjacent cells and include a Sox21a mutant control).*

Sox21a expression was very weak in young flies, and its expression gradually increased with age. Better images were provided and separate color channels for anti-Sox21a staining and NRE>GFP were shown for the enlarged insets (Figure 1). A *Sox21a* mutant control was included, in which anti-Sox21a staining did not detect any obvious signal (Figure 1). Additional specificity test was also provided in Figure 1—figure supplement 1.

The same experiment should then be performed, but after labeling has started and 2-3 EBs generated and Sox21a expression has turned on, DSS treatment should be initiated. The pattern of Sox21a expression 1-2 days later should then be determined and quantitated as above. The authors' model predicts that damage induction will cause a transient decline in Sox21a staining not seen in the control, as the proposed ISC-EB loop is set up and starts to function.

We have taken these suggestions and have redone the regeneration experiment. This time we performed DSS treatment in 7 days old females, when Sox21a has already turned on. In addition, we have modified the damage-induced regeneration protocol. Previously, in order to optimize the damage response and the survival rate of flies, we fed flies with standard food soaked with DSS for 4 consecutive days. To get a more synchronous process of damage or repair, this time we directly fed flies with DSS in sucrose soaked in kimwipe paper for one and half days and dissected the intestine right after the treatment to examine changes at the damage stage, or dissected the intestine 1-2 days later to examine changes at the recovery stage (Figure 6). In agreement with our previous observations, DSS treatment caused a transient decline in Sox21a expression (Figure 6) compared to the control (Figure 6), and the decline of Sox21a expression was accompanied by massive accumulation of progenitor cells, as a result of ISB-EB amplification loop activation. During recovery, Sox21a was dramatically upregulated, which was companied by massive EC differentiation (Figure 6), as Sox21a activation drives EC differentiation. The quantitative measure of Sox21a during the processes is included in Figure 6. We also examined the expression of the Sox21a expression reporter GMR43E09>GFP during the damage and repair process. Consistent with the changes in Sox21a expression, GMR43E09>GFP was transiently downregulated right after the damage and then upregulated 1.5-2 days after the damage (Figure 6—figure supplement 1).

2) The second experiment would be to more clearly document a role for Spi in Sox21a-mediated ISC division. In Zhai's paper, they found that knockdown of Upd2 in the Sox21a mutant background significantly repressed the ISC proliferation, whereas Spi knockdown had only a small effect. The importance of Spi in the ISC-EB feedback loop should be measured quantitatively.

By quantitatively measuring the mitotic index, we found that co-depletion of Spi significantly reduced, although not completely eliminated, the ISC overproliferation phenotype in NRE>Sox21a-RNAi intestine (Figure 5). We believe that this partial effect is likely due to the redundancy of mitogens that participate in the process, such as Pvf2 and Pvf1, both also activate RAS/ MAPK signaling and both are upregulated in *Sox21a* mutant EBs. In our studies, we did not observe upregulation of Upd2 in *Sox21a* mutant EBs, therefore the effect by Upd2 RNAi observed by Zhai et al. could be due to Sox21a-independent mechanisms. As shown in this study, specific depletion of Spi in EBs also significantly reduced the proliferative response of ISCs following DSS treatment. On the other hand, overexpression of Sox21a in EBs eliminated the proliferative response of ISCs (Figure 6). These observations strongly suggest that Spi should be one of the major stimulators of ISC division downstream of Sox21a during homeostatic control of intestinal epithelium.